# Analysis of Polymeric Components in Particulate Matter Using Pyrolysis-Gas Chromatography/Mass Spectrometry

**DOI:** 10.3390/polym14153122

**Published:** 2022-07-31

**Authors:** Eunji Chae, Sung-Seen Choi

**Affiliations:** Department of Chemistry, Sejong University, 209 Neungdong-ro, Gwangjin-gu, Seoul 05006, Korea; codmswl9512@sju.ac.kr

**Keywords:** particulate matter, organic polymeric component, pyrolysis-gas chromatography/mass spectrometry, tire tread wear, asphalt pavement wear

## Abstract

Particulate matters (PMs) such as PM_10_ and PM_2.5_ were collected at a bus stop and were analyzed using pyrolysis-gas chromatography/mass spectrometry to identify organic polymeric materials in them. The major pyrolysis products of the PM samples were isoprene, toluene, styrene, dipentene, and 1-alkenes. The pyrolysis products generated from the PM samples were identified using reference polymeric samples such as common rubbers (natural rubber, butadiene rubber, and styrene-butadiene rubber), common plastics (polyethylene, polypropylene, polystyrene, and poly(ethylene terephthalate)), plant-related components (bark, wood, and leaf), and bitumen. The major sources of the principal polymeric materials in the PM samples were found to be the abrasion of the tire tread and asphalt pavement, plant-related components, and lint from polyester fabric. The particles produced by the abrasion of the tire tread and asphalt pavement on the road were non-exhaustive sources, while the plant-related components and lint from polyester fabric were inflowed from the outside.

## 1. Introduction

Particulate matter (PM) is caused by emitting directly into the atmosphere and by converting from gaseous substances to particles by reaction with other substances in the air [1]. Primary emission sources can be divided into natural sources and anthropogenic sources. PM is generated mainly by anthropogenic factors such as vehicle emissions, industrial fuel combustion, domestic fuel burning, cooking, and biomass combustion [1,2]. PM_2.5_ and PM_10_ refer to particles with an aerodynamic diameter smaller than 2.5 and 10 μm, respectively.

According to a study on PM_2.5_ sources produced for 1990–2014, on average globally, 25% was transportation, 15% was industrial activity, 20% was combustion of household fuel, 22% was unspecified sources of human origin, and 18% was natural dust and sea salt [3]. Research on PM sources from 15 different sampling regions in the Republic of Korea for 2000–2017 reported that motor vehicles, secondary aerosol, combustion and industry, natural source, soil dust, biomass and field burning, and other PM sources were at 28, 34, 14, 2, 7, 11, and 4%, respectively [4].

It is known that PM in the atmosphere is composed of three major components, which are water-soluble ions (SO_4_^2−^, NO_3_^−^, NH_4_^+^, and alkalis and alkaline earth metal cations), carbon (organic carbon and elemental carbon), and heavy metals [1,5]. The water-soluble ions are analyzed by mainly using ion exchange chromatography (IEC) [6,7,8,9]. The carbon components can be divided into organic carbon (OC) and elemental carbon (EC), and thermal methods are used to estimate OC and EC [10,11,12]. Trace metals have been analyzed using X-ray fluorescence, inductively coupled plasma-atomic emission spectroscopy, and inductively coupled plasma-mass spectrometry [5,12,13,14].

Besides the ionic species, OC/EC, and trace metals, PM can also contain various organic components. There can be organic materials such as polyaromatic hydrocarbons (PAHs) and microplastics in PM. PAHs have been analyzed using liquid chromatography and gas chromatography [9,15]. The analysis of microplastics in PM has been primarily focused on classification by color or shape using stereomicroscopy [16,17,18,19,20]. There are various suspended atmospheric microplastics: poly(ethylene terephthalate) (PET), polyethylene (PE), polyamide (PA), polystyrene (PS), polypropylene (PP), and so on. PET- and PA-type plastics are widely used in the textile industry, while PP can be derived from industrial products such as textiles, car fenders, and bottle caps. Plastic products can decompose when they are exposed to sunlight, and their tiny particles may be released into the air.

Tire tread wear particles (TWPs) are generated by friction between the tire tread and road surface. In general, TWPs exist as tire-road wear particles (TRWPs) in the form of TWP encrusted with various mineral particles [2]. Pyrolysis-gas chromatography/mass spectrometry (Py-GC/MS) has been widely used for the analysis of TRWPs [2,21,22,23]. The contribution levels of TRWPs to PM_10_ collected in France, USA, and Japan were assessed using Py-GC/MS, and it was reported that there were TRWPs of 0.14–2.80% in the PM_10_ samples [22]. PM_2.5_ samples collected near houses, parks, schools, and businesses in Europe (London), Japan (Tokyo), and US (Los Angeles) were also analyzed using Py-GC/MS, and it was reported that TRWP contents in the PM_2.5_ samples were 0.11–0.49%, 0.10–0.33%, and 0.10–0.68% in London, Tokyo, and Los Angeles, respectively [2].

Component analysis via pyrolytic technique has been primarily focused on microplastic analysis in environmental samples such as sediment and soil rather than atmospheric samples [24,25,26,27,28]. In this study, PM samples were collected at a bus stop, the organic polymeric components in them were analyzed using Py-GC/MS, and the kinds of polymeric components were determined using the pyrolysis products. Various polymer reference samples such as rubbers, plastics, plant matter, and bitumen were employed to identify the polymeric components in the PM samples. The kinds of polymeric materials in the PM samples were determined by comparing their principal pyrolysis products with those of the references. The sources of the polymeric components in the PM samples were investigated. Py-GC/MS is a useful method for the identification of polymeric materials via the interpretation of the pyrolysis products [29,30], and it is suitable for the analysis of trace organic compound analysis because it can be conducted with a very small sample size.

## 2. Materials and Methods

### 2.1. Materials

#### 2.1.1. Environmental Sample

PM_10_ and PM_2.5_ were collected at a bus stop near Sejong University, Republic of Korea (37°32′58.8″ N 127°04′32.3″ E). The PM_10_ sample was collected in October 2020 (for 12 h), and the PM_2.5_ samples were collected in November 2020 (PM_2.5_(1), for 14 h), February 2021 (PM_2.5_(2), for 24 h), May 2021 (PM_2.5_(3), for 24 h), and August 2021 (PM_2.5_(4), for 24 h). The PM sampling was carried out using a low volume particulate sampler of KMS-4200 (Kemik Co., Seoul, Korea) and a filter with a diameter of 47 mm.

Road dust that accumulated between the curb and road at the bus stop was collected. It was collected by sweeping with a broom and was separated by size using a sieve shaker from Octagon 200 (Endecotts Co., London, UK). TRWP and asphalt wear particles (AWP) were selected using an image analyzer (EGVM 35B, EG Tech. Co., Seoul, Korea).

#### 2.1.2. Reference Samples

SMR CV60, BR01, and SBR1502 were used for NR, BR, and SBR as the reference rubber samples, respectively. Oak tree components of bark, wood, and leaf were employed as the reference plant-related component (PRC) samples. The PRC samples were obtained in Achasan, located in Gwangjin-gu (Korea), where there is no contamination by anthropogenic factors. PE, PP, PS, and PET were used as the reference plastic samples, and they were purchased from Sigma-Aldrich Co. (St. Louis, MI, USA). Bitumen, used as a binder of asphalt pavement, was also employed as the reference sample.

### 2.2. Pyrolysis-Gas Chromatography/Mass Spectrometry (Py-GC/MS)

Py-GC/MS analysis was carried out using a JCI-55 Curie-point pyrolyzer (Japan Analytical Industry Co., Tokyo, Japan) coupled to an Agilent 6890 gas chromatograph equipped with a 5973 mass spectrometer (Agilent Techonology Inc., Santa Clara, CA, USA). All samples were pyrolyzed using a pyrofoil of 590 °C Curie temperature for 10 s under Helium (He) atmosphere. The pyrolysis products were separated through GC, and each separated peak was identified by interpreting its mass spectrum. A DB-5MS capillary column (30 m × 0.32 mm, 0.25 μm film thickness, Agilent Technology Inc., Santa Clara, CA, USA) was used. The injector temperature was 250 °C. Two GC oven temperature programs were used as follows: (1) 30 °C (held for 3 min) to 160 °C (held for 1 min) at 8 °C/min, then to 250 °C (held for 3 min) at 10 °C/min; and (2) 30 °C (held for 3 min) to 50 °C (held for 3 min) at 10 °C/min, then to 180 °C (held for 1 min) at 10 °C/min and raised up to 250 °C (held for 3 min) at 10 °C/min again. The interface temperature of GC to MS was 250 °C. The electron ionization (70 eV) was used to ionize the pyrolysis products. The MS source temperature was 230 °C.

## 3. Results and Discussion

### 3.1. Py-GC/MS Analysis of the PM Samples

Figure 1 and Figure 2 show Py-GC/MS chromatograms of the PM_10_ and PM_2.5_ samples, respectively. Their major pyrolysis products are summarized in Table 1 and Table 2, respectively. The common pyrolysis products detected from the five PM samples were isoprene, toluene, styrene, dipentene, and 1-alkenes. Compared to the PM_10_ sample, there were some different pyrolysis products such as furfural, acetophenone, 4-methylphenol, and benzoic acid in the PM_2.5_ sample. The kinds and abundances of pyrolysis products of the four PM_2.5_ samples were different from each other. The major pyrolysis products of the PM_2.5_(1) sample were nearly the same as those of the PM_10_ sample, whereas the other PM_2.__5_ samples showed some different pyrolysis products, as listed in Table 2. Furfural was detected in the PM_2.5_(2) and PM_2.5_(3) samples. Acetophenone and benzoic acid were not observed in the PM_2.5_(1) sample. The composition of the PM sample should be dependent on the sampling time and weather. The kinds of polymeric materials to generate the principal pyrolysis products were analyzed in comparison with the reference samples of rubbers, plastics, PRCs, and bitumen.

### 3.2. Py-GC/MS Analysis of the Reference Samples

In order to identify the kinds of polymeric materials in the PM samples, principal pyrolysis products of the reference samples such as NR, BR, SBR, PE, PP, PS, PET, bitumen, and PRCs (bark, wood, and leaf) were analyzed. They were pyrolyzed under the same conditions as the PM samples, and their major pyrolysis products were assigned. The principal pyrolysis products of the rubbers (NR, BR, and SBR), plastics (PE, PP, PS, and PET), and bitumen are listed in Table 3, while those of the PRCs are summarized in Table 4.

The principal pyrolysis products of NR were isoprene and dipentene, corresponding to the monomer and dimer of the repeat unit (isoprene), respectively, because NR is a polymer of isoprene, and the key pyrolysis products of isoprene and dipentene are the monomer and dimer of the repeat unit [31,32,33]. The isoprene and dipentene were detected as main pyrolysis products in the PM samples. This implies that there was an NR component in the PM samples. NR should originate from tire tread wear particles. In general, bus tire treads are mainly made of NR [34,35,36,37,38]. Hence, the source of the NR should be bus tire tread wear. Abrasion of tire tread occurs mainly during the start and stop of a vehicle rather than during stable driving. The principal pyrolysis products of BR are 1,3-butadiene and 4-vinylcyclohexene (VCH, dimer of 1,3-butadiene), but there was no VCH in the PM samples. This indicates that there was no BR component in the PM samples. There are various principal pyrolysis products of SBR, such as 1,3-butadiene, VCH, styrene, 2-phenylpropene (2-PP), 3-phenylcyclopentene (3-PCP), and 4-phenylcyclohexene (4-PCH) [39,40,41,42]. If there were SBR components in the PM samples, some of them must have been detected. Styrene was clearly observed in the Py-GC/MS chromatograms of the PM samples, but most of the others were not detected. This implies that there was no SBR component in the PM samples and that the source of the styrene was not SBR.

It is well known that the principal pyrolysis products of PE are alkanes, 1-alkenes, and alkadienes [26]. In Py-GC/MS chromatograms of the PM samples, 1-alkenes were clearly detected, but alkanes and alkadienes were rarely observed. Hence, the source of 1-alkenes detected in the PM samples was not PE. 2-Methyl-1-pentene, 2,4-dimethyl-1-heptene, and 2,4,6-trimethyl-1-nonene are typical pyrolysis products of PP, but they were not detected in the PM samples. This indicates that there was no PP component in the PM samples. Styrene is one of the major pyrolysis products of PS, and it was observed in the PM samples, but 2-PP and biphenyl, as the major pyrolysis products of PS, were not detected in the PM samples. Thus, PS was not the source of styrene in the PM samples. Acetophenone and benzoic acid were observed in the PM_2.5_(2) and PM_2.5_(3) samples, and they are principal pyrolysis products of PET [43,44]. This implies that there was a PET component in the PM_2.5_(2) and PM_2.5_(3) samples.

Bitumen is widely used as a binder of asphalt pavement. The vomponents of bitumen vary widely, and they are different according to the suppliers. The principal pyrolysis products of bitumen were 1,3-butadiene, styrene, phenol, 1-ethenyl-4-methylbenzene, indene, 1-undecene, 1-dodecene, and 4-(1-methylethyl)-phenol, as listed in Table 3. Among them, the pyrolysis products that matched with the PM samples were styrene and 1-alkenes such as 1-undecene and 1-dodecene. Hence, it can be concluded that one of the styrene sources should be bitumen and that one of the 1-alkene sources should also be bitumen.

The principal pyrolysis products of the PRCs varied according to the kinds of components, such as bark, wood, and leaf, as listed in Table 4. The kinds of principal pyrolysis products of the two leaf samples were nearly the same, but the relative abundances were different from each other. One of the key pyrolysis products of PRCs is furfural produced from cellulose [45,46]. Furfural was abundantly observed in all the PRC samples. 1-Alkenes such as 1-undecene and 1-dodecene were also detected in the PRC samples. 1-Alkenes should be pyrolysis products of waxes, and it is known that there is plant epicuticular wax in the surface layer [47,48]. 1-Alkenes were observed in all the Py-GC/MS chromatograms of the PM samples, and furfural was detected in those of the PM_2.5_(2) and PM_2.5_(3) ones. Since furfural is the pyrolysis product of the wood component, the analytical results indicate that the PM_2.5_(2) and PM_2.5_(3) samples contained the wood component. The PM samples might contain some bark or leaf component because plant epicuticular wax generally exists in the surface layer of bark and leaf.

### 3.3. Py-GC/MS Analysis of the TRWP and AWP Collected near the Bus Stop

Single TRWP and AWP collected near the bus stop were pyrolyzed, and their Py-GC/MS chromatograms are shown in Figure 3 and Figure 4, respectively. The TRWP chromatogram clearly showed the key pyrolysis products of NR (isoprene and dipentene), and styrene was also observed, but VCH was not detected. If the TRWP was a debris of an SBR or NR/BR compound, the chromatogram must show VCH as one of the major pyrolysis products because VCH is the key pyrolysis product of BR and SBR. Hence, the TRWP was a particle of an NR compound, and it should come from the abrasion of bus tire tread composed of NR. Thus, it can be concluded that the isoprene and dipentene detected in the PM samples originated from abrasion of bus tire tread.

The Py-GC/MS chromatogram of the AWP showed various pyrolysis products (Figure 4). The major pyrolysis products were 1-alkenes, toluene, and styrene. The 1-alkenes detected in the PM samples should come from the AWPs. Since styrene was also one of the major pyrolysis products of the AWP, its detection in the TRWP should come from the AWP. During the abrasion process of the bus tire tread, friction between the tire tread and asphalt pavement occurs, and a few small AWPs can be attached to the TWP surface.

### 3.4. Sources of the Organic Polymeric Components in the PM Samples

The sources of the organic polymeric components in the PM_10_ sample are summarized in Figure 5. Isoprene and dipentene are the key pyrolysis products of NR, which is the main component of bus tire tread compounds. Hence, isoprene and dipentene originated from TWPs of buses. PRCs could also produce isoprene and dipentene as pyrolysis products, but their abundances were very small, so the contribution level of the PRCs in producing isoprene and dipentene could be negligible. Styrene was one of the major pyrolysis products formed from AWP, and it was also one of the main pyrolysis products of PS and SBR. However, styrene dimer and trimer and VCH were not detected in the PM samples. Thus, styrene should originate from AWP. 1-Alkenes are key pyrolysis products of wax and one of the major pyrolysis products of bitumen. PRCs also have epicuticular wax. Hence, the sources of 1-alkenes should be PRCs and AWP.

The sources of the organic polymeric components in the PM_2.5_ samples are described in Figure 6. The PM_2.5_ samples also showed isoprene, dipentene, styrene, and 1-alkenes as the pyrolysis products. Hence, like the PM_10_ sample, the PM_2.5_ samples contained TWPs, AWPs, and PRCs. Additionally, furfural, acetophenone, and benzoic acid were observed in the PM_2.5_ samples. Acetophenone and benzoic acid are major pyrolysis products of PET, which can originate from the lint of polyester (PET) fabric/clothes. The detection of furfural can provide evidence that the sample contains a wood component.

Figure 7 shows the kinds of organic polymeric materials in the PM samples collected at the bus stop, and the key pyrolysis products for identifying the kinds of polymeric materials by using Py-GC/MS. In summary, there are TRWPs, AWPs, PRCs, and polyester fabric (lint of clothes) as PM near bus stops on asphalt pavement roads. The kinds and abundances of polymeric components in the PM samples can be dependent on the sampling site, time, road conditions, and traffic volumes.

## 4. Conclusions

The kinds of polymeric components in PM samples were identified by interpreting their pyrolysis products. In order to clearly identify the kinds of polymeric materials in the PM samples, reference samples such as NR, BR, SBR, PE, PP, PS, PET, bitumen, and PRCs (bark, wood, and leaf) were employed, and their pyrolysis products were analyzed. The major pyrolysis products generated from the PM samples were isoprene, toluene, styrene, dipentene, and 1-alkenes. The main polymeric components found in the PM samples were wear particles of tire tread and asphalt pavement, plant-related components, and the lint of polyester fabric. Compared to the PM_10_ sample, there were some different pyrolysis products such as furfural, acetophenone, 4-methylphenol, and benzoic acid in the PM_2.5_ samples. Furfural is one of the major pyrolysis products of plant-related components, while acetophenone and benzoic acid are the principal pyrolysis products of PET. The kinds and abundances of polymeric components in PM samples can be dependent on the sampling site, time, road conditions, and traffic volumes. The analysis of various polymeric components in PM samples can be useful for understanding various sources of PM.

## Figures and Tables

**Figure 1 polymers-14-03122-f001:**
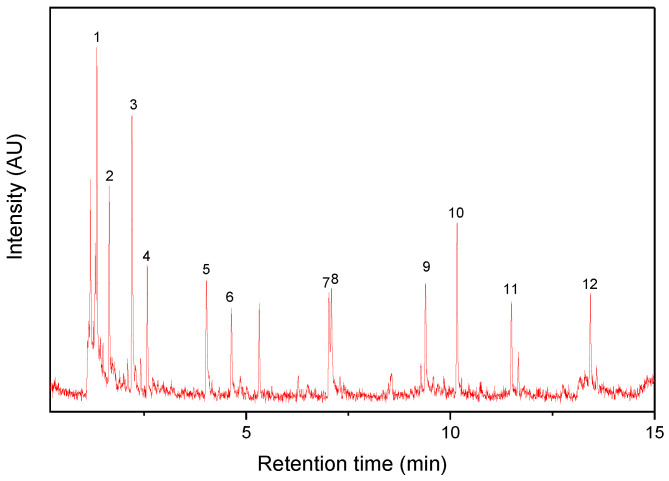
Py-GC/MS chromatogram of the PM_10_ sample.

**Figure 2 polymers-14-03122-f002:**
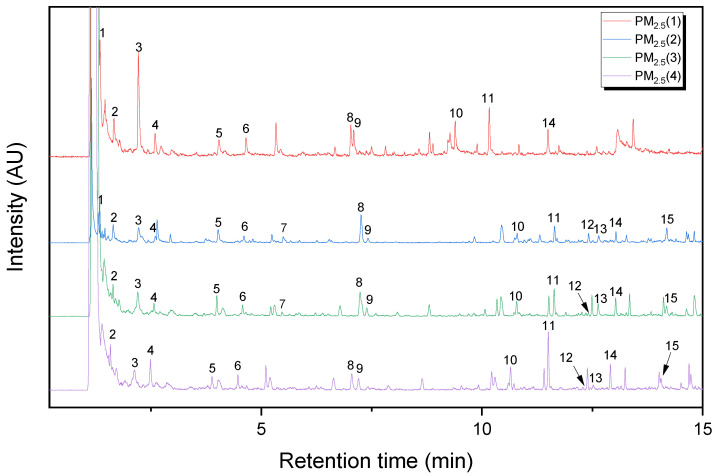
Py-GC/MS chromatograms of the PM_2.5_(1), PM_2.5_(2), PM_2.5_(3), and PM_2.5_(4) samples.

**Figure 3 polymers-14-03122-f003:**
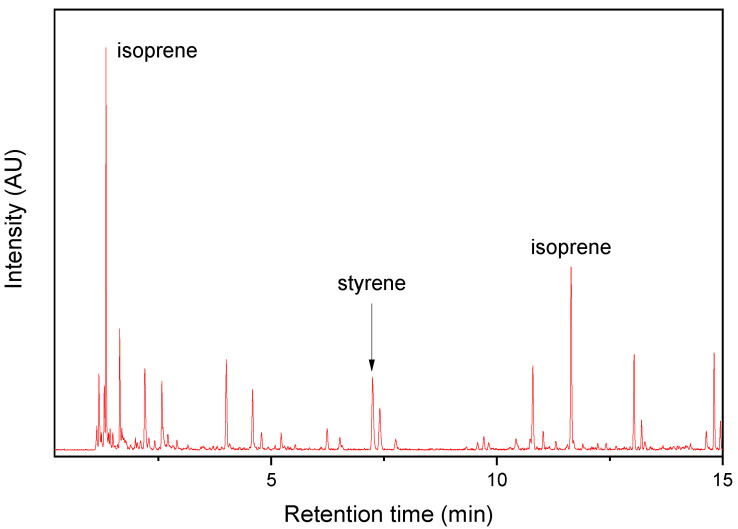
Py-GC/MS chromatogram of the TRWP collected near the PM sampling site.

**Figure 4 polymers-14-03122-f004:**
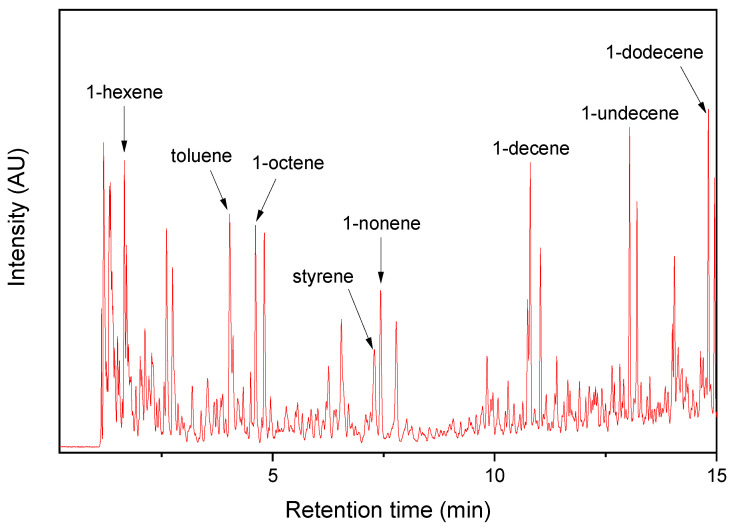
Py-GC/MS chromatogram of the AWP collected near the PM sampling site.

**Figure 5 polymers-14-03122-f005:**
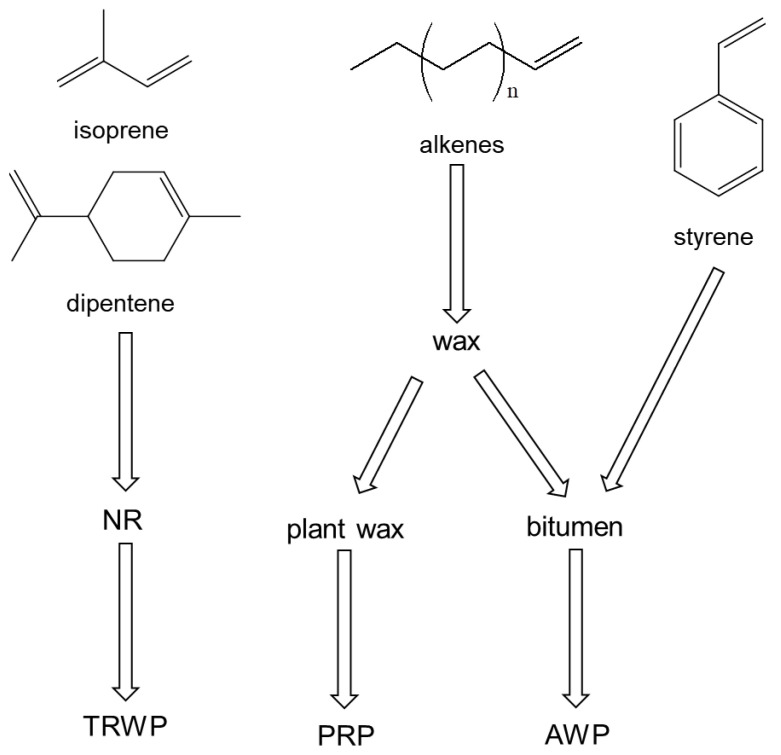
Sources of the principal pyrolysis products obtained from the PM_10_ sample.

**Figure 6 polymers-14-03122-f006:**
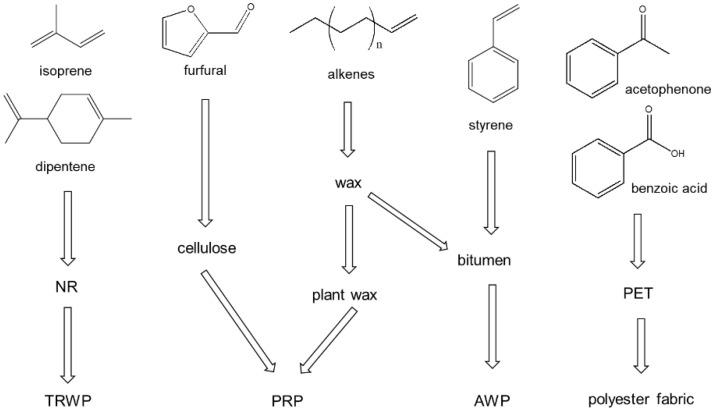
Sources of the principal pyrolysis products obtained from the PM_2.5_ samples.

**Figure 7 polymers-14-03122-f007:**
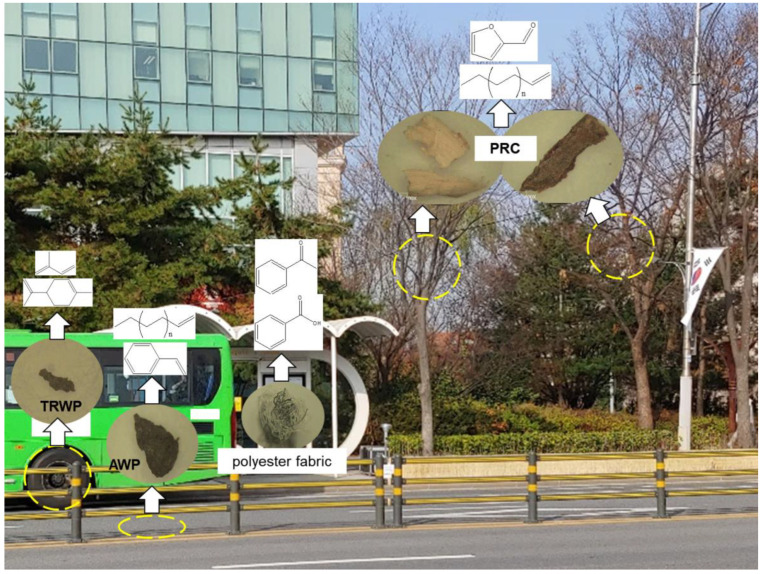
PM sources produced near the bus stop, and the key pyrolysis products for identifying polymeric materials using Py-GC/MS. The yellow circles indicate potential PM sources.

**Table 1 polymers-14-03122-t001:** Peak areas and relative intensity ratios (the reference: 1-undecene) of the major pyrolysis products detected in the PM_10_ sample.

Peak No.	Product	Peak Area (10^3^) (Intensity Ratio)
1	isoprene	668 (1.70)
2	1-hexene	526 (1.33)
3	benzene	1339 (3.40)
4	1-heptene	511 (1.30)
5	toluene	520 (1.32)
6	1-octene	316 (0.80)
7	styrene	614 (1.56)
8	1-nonene	606 (1.54)
9	1-decene	440 (1.12)
10	dipentene	746 (1.89)
11	1-undecene	394 (1.00)
12	1-dodecene	287 (0.73)

**Table 2 polymers-14-03122-t002:** Peak areas and relative intensity ratios (the reference: 1-undecene) of the major pyrolysis products detected in the PM_2.5_ samples.

Peak No.	Pyrolysis Product	Peak Area (10^3^) (Intensity Ratio)			
		PM_2.5_(1)	PM_2.5_(2)	PM_2.5_(3)	PM_2.5_(4)
1	isoprene	697 (0.83)	339 (1.09)	438 (1.08)	602 (0.76)
2	1-hexene	591 (0.70)	trace	413 (1.01)	531 (0.67)
3	benzene	4121 (4.92)	1395 (4.47)	762 (1.87)	595 (0.75)
4	1-heptene	734 (0.88)	167 (0.53)	273 (0.67)	1065 (1.34)
5	toluene	664 (0.79)	548 (1.76)	737 (1.81)	498 (0.63)
6	1-octene	551 (0.66)	188 (0.60)	288 (0.71)	457 (0.58)
7	furfural	---	196 (0.63)	46 (0.11)	---
8	styrene	1381 (1.65)	1540 (4.94)	1768 (4.34)	792 (1.00)
9	1-nonene	846 (1.01)	224 (0.72)	394 (0.97)	515 (0.65)
10	1-decene	873 (1.04)	254 (0.82)	444 (1.09)	742 (0.94)
11	dipentene	1479 (1.76)	529 (1.70)	880 (2.16)	1991 (2.51)
12	acetophenone	---	316 (1.01)	51 (0.13)	trace
13	4-methylphenol	---	263 (0.84)	92 (0.23)	136 (0.17)
14	1-undecene	838 (1.00)	312 (1.00)	407 (1.00)	792 (1.00)
15	benzoic acid	---	561 (1.80)	506 (1.24)	174 (0.22)

**Table 3 polymers-14-03122-t003:** Principal pyrolysis products of the reference organic polymeric samples.

Polymer	Principal Pyrolysis Products
NR	isoprene, dipentene
BR	1,3-butadiene, 4-vinylcyclohexene (VCH)
SBR	1,3-butadiene, VCH, styrene, 2-phenylpropene (2-PP), 3-phenylcyclopentene (3-PCH), 4-phenylcyclohexene (4-PCH)
PE	alkanes, 1-alkenes, alkadienes
PP	2-methyl-1-pentene, 2,4-dimethyl-1-heptene, 2,4,6-trimethyl-1-nonene
PS	styrene, 2-PP, biphenyl
PET	benzene, acetophenone, vinyl benzoate, benzoic acid, 4-methylbenzoic acid
Bitumen	1,3-butadiene, styrene, phenol, 1-ethenyl-4-methylbenzene, indene, 1-undecene, 1-dodecene, 4-(1-methylethyl)-phenol

**Table 4 polymers-14-03122-t004:** Relative intensity ratios (the reference: 2-methoxy-4-methylphenol) of principal pyrolysis products produced from the oak tree components.

Pyrolysis Product	Bark	Wood	Leaf-1	Leaf-2
furfural	0.27	1.30	0.80	0.94
styrene	0.14	0.03	0.25	0.12
1-decene	0.09	---	---	---
2,2-diethyl-3-methyl-oxazolidine	0.12	1.64	0.49	0.87
dipentene	0.07	0.16	0.22	0.11
4-methylphenol	0.31	0.13	0.73	0.35
2-methoxyphenol	0.69	0.88	0.72	0.62
1-undecene	0.08	---	---	---
maltol	---	0.17	0.17	0.19
2-methoxy-4-methylphenol	1.00	1.00	1.00	1.00
1-dodecene	0.09	---	---	---

## Data Availability

The data presented in this study are available on request from the corresponding author.

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
