# Peer review of "Analysis of Polymeric Components in Particulate Matter Using Pyrolysis-Gas Chromatography/Mass Spectrometry"

_polymers, 2022, doi:10.3390/polym14153122_

Round 1

Reviewer 1 Report

The authors have studied the samples of particulate matter collected at bus stations in Korea using pyrolysis of the polymers and the subsequent GC-MS analysis of the destruction products. The provided paper reports the everyday routine work of the chromatographic lab. So, the scientific soundness and level of the provided study are low. Therefore, I guess the manuscript is not a topic of a high-quality scientific journal such as Polymers, albeit it is well-prepared using good English and style. The authors should address the following issues to improve their work:

1. The Abstract should be rewritten to demonstrate the primary outcomes of the performed study and highlight its novelty, significance of content, and scientific soundness or practical value.

2. The authors should clearly describe how pyrolysis can facilitate the identification of polymers.

3. Lines 36-45: The information in this paragraph is trivial, banal, and unrelated to the paper’s theme. Therefore, I recommend removing this paragraph.

4. Line 37: Please, check the anions' spelling.

5. There is no aim in the introductory part. The authors should clearly formulate the primary goal of the implemented study. Furthermore, they should rewrite the Conclusions to highlight the paper’s scientific and practical value.

6. The authors, throughout the manuscript, use a lot of acronyms that hinder the article’s reading. Please, shorten the number of acronyms in the manuscript.

7. Lines 105-113: Check degrees Celcius spelling. Check also the correctness of the chromatographic column characteristics.

Author Response

Q1. The Abstract should be rewritten to demonstrate the primary outcomes of the performed study and highlight its novelty, significance of content, and scientific soundness or practical value.

A1. The Abstract was rewritten, and the corrections were marked in blue.

Q2. The authors should clearly describe how pyrolysis can facilitate the identification of polymers.

A2. A certain polymer generates its specific pyrolysis products (key porolysis products). For example, isoprene and dipentene are key porolysis products of natural rubber (NR) because NR is a polymer of isoprene. In this study, the reference samples such as NR, BR, SBR, PE, PP, PS, PET, bitumen, and PRCs (bark, wood, and leaf) were employed and their key porolysis products were analyzed first to clearly identify the kinds of the polymeric materials in the PM samples. In the section “3.2. Py-GC/MS analysis of the reference samples”, the sentence “Principal pyrolysis products of NR were isoprene and dipentene corresponding to the monomer and dimer of NR, respectively [31-33].” was changed to “Principal pyrolysis products of NR were isoprene and dipentene corresponding to the monomer and dimer of NR, respectively, because NR is a polymer of isoprene and the key pyrolysis products of isoprene and dipentene are monomer and dimer of the repeat unit (isoprene) [31-33].” to clearly describe the key pyrolysis products. The sentence “Principal pyrolysis products of BR are 1,3-butadiene and 4-vinylcyclohexene (VCH),” was changed to “Principal pyrolysis products of BR are 1,3-butadiene and 4-vinylcyclohexene (VCH, dimer of 1,3-butadiene),” to clearly describe the key pyrolysis products.

Q3. Lines 36-45: The information in this paragraph is trivial, banal, and unrelated to the paper’s theme. Therefore, I recommend removing this paragraph.

A3. In this paragraph, general components of PM and their analytical methods were described. For the sake of readers’ understanding for PM and related analytical techniques, we think that this paragraph is needed. The paragraph size was reduced by changing the sentence “The carbon components can be divided into organic carbon (OC) and elemental carbon (EC). OC is a primary pollutant directly emitted during incomplete combustion of carbon-containing fuels, while EC is mainly soot produced by combustion [10]. Thermal methods are used to estimate OC and EC [11,12].” to “The carbon components can be divided into organic carbon (OC) and elemental carbon (EC), and thermal methods are used to estimate OC and EC [10-12].”.

Q4. Line 37: Please, check the anions' spelling.

A4. In the original manuscript, symbol of the anions and cation was correctly written as the upper character. We think that these are errors generated when transferring the original manuscript to the Polymers’ Format, and the misspelling was corrected.

Q5. There is no aim in the introductory part. The authors should clearly formulate the primary goal of the implemented study. Furthermore, they should rewrite the Conclusions to highlight the paper’s scientific and practical value.

A5. The aim was identification of kinds of the polymeric materials in PM samples. The aim was described in the Introduction part. The sentence “In this study, PM samples were collected at a bus stop and the organic polymeric components in them were analyzed using Py-GC/MS.” was changed to “In this study, PM samples were collected at a bus stop, the organic polymeric components in them were analyzed using Py-GC/MS, and kinds of the polymeric components were determined using the pyrolysis products.”, and the sentences in the last paragraph in the Introduction part were rearranged to clear the aim. The conclusion part was rewritten and the corrections were marked in blue.

Q6. The authors, throughout the manuscript, use a lot of acronyms that hinder the article’s reading. Please, shorten the number of acronyms in the manuscript.

A6. Acronyms for the reference materials in the Abstract were deleted. Acronyms for the analytical instruments such as XRF, ICP-AES, ICP-MS, LC, GC in the Introduction part were deleted. Full names for the reference materials in the section “2.1.2. Reference samples” were deleted.

Q7. Lines 105-113: Check degrees Celcius spelling. Check also the correctness of the chromatographic column characteristics.

A7. In the original manuscript, symbol of Celcius temperature was correctly written and the chromatographic column characteristics was correctly written. We think that these are errors generated when transferring the original manuscript to the Polymers’ Format. For the symbol of Celcius temperature, the upper script was not correctly applied. For the chromatographic column characteristics, the multiplication symbol ‘´’ was omitted. The misspelling was corrected.

Reviewer 2 Report

The manuscript by these authors deals with the pyrolysis technique to identify the polymeric components in the particulate. The topic is of great importance, the scientific level of the manuscript is not so high, anyway it could be published in Polymers after e revision, by taking into account the suggestions reported in the attached report. The text is written in good English but the manuscript suffers from a high percentage (25%) of self-citation that must be reduced.

Author Response

Q1. About the misspelling of ions and temperatures.

A1. In the original manuscript, symbols of the ions and temperature was correctly written as the lower and upper characters. We think that these are errors generated when transferring the original manuscript to the Polymers’ Format. The misspelling was corrected.

Q2. About unis of y-axis in the chromatograms.

A2. Arbitrary unit (AU) was added in the chromatograms. Peak areas in Figures 1 and 2 were listed in Tables 1 and 2, respectively.

Q3. I found this figure not useful and in my opinion can be removed from the text.

A3. For the sake of readers’ understanding for identification and sources of the polymeric materials in PM, we think Figure 7 is needed.

Q4. The text is written in good English but the manuscript suffers from a high percentage (25%) of self-citation that must be reduced.

A4. References 33, 39, and 42 were changed other references.

Reviewer 3 Report

The present work deals with the Analysis of polymeric components in particulate matter using pyrolysis-gas chromatography/mass spectrometry. The article can be accepted after addressing minor comments.

Abstract

1. Significant numerical results can be included in the abstract and the graphical abstract need to be improved

2. List of abbreviations can be included after the abstract and a proof reading with a native speaker can be included.

Introduction

4. Recent references related to the manuscript can be included. Lengthy sections can be split up into paragraphs.

5. Check and refer these studies to improve the introduction section, Forecasting of future greenhouse gas emission trajectory for India using energy and economic indexes with various metaheuristic algorithms; Effects of hybrid nanoparticle additives in n-butanol/waste plastic oil/diesel blends on combustion, particulate and gaseous emissions from diesel engine evaluated with entropy..; Energy, exergy, sustainability and economic analysis of waste tire pyrolysis oil blends with different nanoparticle additives in spark ignition engine

Materials and methods

6. Abbreviations can be avoided in subheadings and properties table can be enhanced

Results and discussion

7. Each part can be discussed with deviations in numerical values. Like percentage decrease or percentage increase.

8. Please check and refer these studies for uncertainty, Experimental assessment of diverse diesel engine characteristics fueled with an oxygenated fuel added lemon peel biodiesel blends; Investigation on the effect of cottonseed oil blended with different percentages of octanol and suspended MWCNT nanoparticles on diesel engine characteristics.

9. Please improve the overall R and D section, please add previous studies to support your claims.

Conclusion

10. Conclusion section can be discussed with including numerical values and compare with the relevant work.

11. Future scope of the work can be included in the conclusion.

Author Response

Q1. Significant numerical results can be included in the abstract and the graphical abstract need to be improved

A1. This manuscript is about qualification of kinds of the polymeric components in PM samples not quantification. There is no numerical data except for the peak intensities in this study. I know that the graphical abstract is not essential. The Abstract was rewritten, and the corrections were marked in blue.

Q2. List of abbreviations can be included after the abstract and a proof reading with a native speaker can be included.

A2. Abbreviations in the Abstract part were deleted except for PM. The Abstract was rewritten, and the corrections were marked in blue.

Q3. Recent references related to the manuscript can be included. Lengthy sections can be split up into paragraphs.

A3. 30 References were included in the Introduction part. And 19 References were also included in the Results and Discussion part. The first paragraph in the Introduction part was divided into two paragraphs.

Q4. Check and refer these studies to improve the introduction section, Forecasting of future greenhouse gas emission trajectory for India using energy and economic indexes with various metaheuristic algorithms; Effects of hybrid nanoparticle additives in n-butanol/waste plastic oil/diesel blends on combustion, particulate and gaseous emissions from diesel engine evaluated with entropy..; Energy, exergy, sustainability and economic analysis of waste tire pyrolysis oil blends with different nanoparticle additives in spark ignition engine

A4. Thanks for your comments. This manuscript is not directly related to greenhouse gas emission or energy, exergy, sustainability, and economic analysis of waste tire pyrolysis oil. This manuscript is about identification of kinds of the polymeric components in PM samples.

Q5. Abbreviations can be avoided in subheadings and properties table can be enhanced.

A5. Abbreviations in the Abstract part were deleted except for PM. Abbreviations of XRF, ICP-AES, and ICP-MS in the second paragraph and LC and GC in the third paragraph of the Introduction part were also deleted. Full names of the rubbers and plastics in the section “2.1.2. Reference samples” were deleted because their full names were described in the Introduction part.

Q6. Each part can be discussed with deviations in numerical values. Like percentage decrease or percentage increase.

A6. In this study, kinds of the polymeric components in the PM samples were qualitatively analyzed. Quantitative analysis was not performed. As you know, there is no numerical values except for the peak intensities in this manuscript.

Q7. Please check and refer these studies for uncertainty, Experimental assessment of diverse diesel engine characteristics fueled with an oxygenated fuel added lemon peel biodiesel blends; Investigation on the effect of cottonseed oil blended with different percentages of octanol and suspended MWCNT nanoparticles on diesel engine characteristics.

A7. In this study, kinds of the polymeric components in the PM samples were analyzed. The reference samples such as NR, BR, SBR, PE, PP, PS, PET, bitumen, and PRCs (bark, wood, and leaf) were employed and their key porolysis products were analyzed first to clearly identify the kinds of the polymeric materials in the PM samples. A certain polymer generates its specific pyrolysis products (key porolysis products). For example, isoprene and dipentene are key porolysis products of natural rubber (NR) because NR is a polymer of isoprene. Kinds of the polymeric materials were determined by detection of the key pyrolysis products.

Q8. Please improve the overall R and D section, please add previous studies to support your claims.

A8. In the Results and Discussion part, 19 references were cited to explain the experimental results. To clearly describe the key pyrolysis products, in the section “3.2. Py-GC/MS analysis of the reference samples”, the sentence “Principal pyrolysis products of NR were isoprene and dipentene corresponding to the monomer and dimer of NR, respectively [31-33].” was changed to “Principal pyrolysis products of NR were isoprene and dipentene corresponding to the monomer and dimer of NR, respectively, because NR is a polymer of isoprene and the key pyrolysis products of isoprene and dipentene are monomer and dimer of the repeat unit (isoprene) [31-33].” and the sentence “Principal pyrolysis products of BR are 1,3-butadiene and 4-vinylcyclohexene (VCH),” was changed to “Principal pyrolysis products of BR are 1,3-butadiene and 4-vinylcyclohexene (VCH, dimer of 1,3-butadiene),” to clearly describe the key pyrolysis products.

Q9. Conclusion section can be discussed with including numerical values and compare with the relevant work.

A9. In this study, kinds of the polymeric components in the PM samples were qualitatively analyzed. Since quantitative analysis was not performed, there is no numerical data except for the peak intensities in this manuscript. The conclusion part was rewritten, and the corrections were marked in blue.

Q10. Future scope of the work can be included in the conclusion.

A10. The conclusion part was rewritten, and the corrections were marked in blue.

Round 2

Reviewer 1 Report

The authors have addressed most of my recent concerns. Therefore, I recommend the revised manuscript be accepted for publishing in Polymers in the current version.

Author Response

Q. The authors have addressed most of my recent concerns. Therefore, I recommend the revised manuscript be accepted for publishing in Polymers in the current version.

A. Thanks for your kindness.

Reviewer 3 Report

 1. Extensive editing of English language and style required.

2. Need to improve the abstract, numerical values shoud be added.

3. Loads of Abbreviations throughout the manuscript, need to reduce and add in nomenclature.

4. Explain the procedure of GC/MS analysis.

5. Redraw Fig. 5. and 6, which lacks clarity or inlcude block dig.

6. Add high-resolution image for Fig.7 

7. Conclusion should be written point-wise and add future recommendation for the investigation.

Author Response

Q1. Extensive editing of English language and style required.

A1. This manuscript was reviewed again. Then corrections were marked in blue.

Q2. Need to improve the abstract, numerical values shoud be added.

A2. This manuscript is about qualification of kinds of the polymeric components in PM samples not quantification. There is no numerical data except for the peak intensities in this study. The sentence “Major pyrolysis products of the PM samples were isoprene, toluene, styrene, dipentene, and 1-alkenes.” was added in the Abstract part. The sentence “Wear particles related to abrasion of tire tread and asphalt pavement were non-exhaustive sources produced from the road,” was changed to “Particles produced by abrasion of the tire tread and asphalt pavement on the road were non-exhaustive sources,”.

Q3. Loads of Abbreviations throughout the manuscript, need to reduce and add in nomenclature.

A3. Full names of the abbreviations were described first. In this study, many reference samples were used and their names were described as their proper abbreviations.

Q4. Explain the procedure of GC/MS analysis.

A4. The GC/MS analysis conditions were already described in the section “2.2. Pyrolysis-gas chromatography/mass spectrometry (Py-GC/MS)” in detail. The sentence “The pyrolysis products were separated through GC, and each separated peak was identified by interpreting its mass spectrum.” was added in the section.

Q5. Redraw Fig. 5. and 6, which lacks clarity or inlcude block dig.

A5. Figures 5 and 6 were redrawn.

Q6. Add high-resolution image for Fig.7 

A6. Figure 7 was redrawn.

Q7. Conclusion should be written point-wise and add future recommendation for the investigation.

A7. Analysis of various polymeric components in PM samples can be useful for understanding various sources of PM.